# An Improved Animal Model of Multiple Myeloma Bone Disease

**DOI:** 10.3390/cancers13174277

**Published:** 2021-08-25

**Authors:** Syed Hassan Mehdi, Carol A Morris, Jung Ae Lee, Donghoon Yoon

**Affiliations:** 1Myeloma Center, University of Arkansas for Medical Sciences, Little Rock, AR 72205, USA; SHMehdi@uams.edu; 2Graduate Program in Interdisciplinary Biomedical Sciences, University of Arkansas for Medical Sciences, Little Rock, AR 72205, USA; CAMorris@uams.edu; 3Department of Population and Quantitative Health Sciences, University of Massachusetts Medical School, Worcester, MA 01605, USA; JungAe.Lee@umassmed.edu

**Keywords:** murine model, multiple myeloma bone disease, severe osteolytic lesions

## Abstract

**Simple Summary:**

Multiple myeloma is a plasma cell cancer involving bone destruction and is considered an incurable disease despite significant improvements in therapeutic strategies. During myeloma progression, over 90% of patients develop a bone disease that causes patient injury and death. There are limited animal models available to demonstrate multiple myeloma bone disease (MMBD). The current study identifies and validates the newly developed MMBD models with uniformity of tumor burden and severe bone lesions. This model will help study the biology of MMBD and serve as a valuable tool for screening therapeutic candidates by monitoring their response to disease progression.

**Abstract:**

Multiple myeloma (MM) is a plasma cell malignancy that causes an accumulation of terminally differentiated monoclonal plasma cells in the bone marrow, accompanied by multiple myeloma bone disease (MMBD). MM animal models have been developed and enable to interrogate the mechanism of MM tumorigenesis. However, these models demonstrate little or no evidence of MMBD. We try to establish the MMBD model with severe bone lesions and easily accessible MM progression. 1 × 10^6^ luciferase-expressing 5TGM1 cells were injected into 8–12 week-old NOD SCID gamma mouse (NSG) and C57BL/KaLwRij mouse via the tail vein. Myeloma progression was assessed weekly via in vivo bioluminescence (BL) imaging using IVIS-200. The spine and femur/tibia were extracted and scanned by the micro-computer tomography for bone histo-morphometric analyses at the postmortem. The median survivals were 56 days in NSG while 44.5 days in C57BL/KaLwRij agreed with the BL imaging results. Histomorphic and DEXA analyses demonstrated that NSG mice have severe bone resorption that occurred at the lumbar spine but no significance at the femur compared to C57BL/KaLwRij mice. Based on these, we conclude that the systemic 5TGM1 injected NSG mouse slowly progresses myeloma and develops more severe MMBD than the C57BL/KaLwRij model.

## 1. Introduction

Multiple myeloma (MM) is the second most common hematological threat in the USA, characterized by the accumulation of terminally differentiated monoclonal plasma cells (PCs) in the bone marrow (BM) [1,2]. MM is characterized by hypercalcemia, renal failure, anemia, and lytic bone lesions. Over 90% of patients develop one or more bone lesions during the disease, called MM-associated bone disease (MMBD) [3,4,5,6,7]. MMBD develops when MM cells disrupt the balance of bone absorption and formation, resulting in activation of osteolytic devastation [8]. MM progression and osteolytic lesion development are highly linked, demonstrating the interaction of myeloma cells with the bone marrow microenvironment. Furthermore, osteoimmunology describes interdisciplinary mechanisms of bone and immune cells in both normal and pathophysiology [9]. A number of factors, including Receptor Activator of NF-κB (RANK), RANK ligand, osteoprotegerin, play roles in osteolytic lesions and MM progression. Some of these factors are produced from various immune cells. These findings suggest that crosstalk between the immune system and bone cells may affect cancer growth [10].

Various murine models have been developed to study the biology of MM. These models describe the essential tool to explore and envisage the efficacy of innovative therapeutic approaches [11]. The 5TGM1 transplanting C57BL/KaLwRij mouse via the tail vein develops MMBD features close to human MMBD [12,13,14,15,16]. This model has a short latency period to develop MM and significant osteolytic bone lesions [12]. However, we found significant variations of MMBD levels per individual mouse [17,18,19]. The mouse/human myeloma cells (5TGM1, U266, or JJN3) were intratibially injected into severe combined immunodeficient (SCID) or non-obese diabetic (NOD)/SCID. They developed MMBD but progression was limited to the tibia where tumor cells were injected [20,21,22]. The human GFP-expressing RPMI-8226/S MM cells were intravenously injected into NOD/SCID mice and demonstrated that the human MM cells infiltrate into mouse bone and sequentially develop MMBD [23]. The primary human MM cells were orthotopically injected into human fetal or rabbit bones implanted SCID mice and developed bone lesions on implanted bones [24,25,26,27]. Later, the NOD/SCID-GAMMA (NSG) mice were used to study various levels of BM infiltration and overall survival [28,29,30,31], which also have limited information on the developments of MMBD and osteolytic lesions [32,33,34]. While the orthotopic transplanting models showed extensive MMBD at the injected bone site, systemic injection models have various tumor burden and MMBD levels. In addition, the early/quantifiable detection of myeloma and myeloma progression in the murine models will be useful in the therapeutic evaluation.

In the current study, we establish the MMBD model with severe bone lesions and easily accessible MM progression. In addition, we evaluated the times of disease detection, the progression of the tumor, and the extent of bone lesions in systemically injected 5TGM1 (5TGM1-*Luc*) cells in the NSG or C57BL/KaLwRij mice.

## 2. Materials and Methods

### 2.1. Cell Culture

5TGM1 cells were received from the University of Texas Health Science Center, San Antonio, and maintained in RPMI 1640 medium + 15% fetal bovine serum (FBS) + 1 × penicillin/streptomycin + 1 × Glutmax (Gibco, Life Technologies, Carlsbad, CA, USA) at 37 °C in an atmosphere of 5% CO_2_ /95% air. 5TGM1-Luc cells were maintained in a range of 0.5–2 × 10^6^ cells/mL. Before the transplant, cells were washed with PBS three times and counted by Cellometer Mini (Nexcelom, Lawrence, MA, USA) using the trypan blue exclusion method.

### 2.2. Intravenous Inoculation of 5TGM1-Luc Cells

NSG and C57BL/KaLwRij mice were housed and bred at the University of Arkansas for Medical Sciences (UAMS) Animal Facility. All animal procedures were reviewed and approved by the UAMS IACUC and were conducted as per the National Institutes of Health (NIH) Guide for the Care and Use of Laboratory Animals. 1 × 10^6^ cells (>90% viability) in 100 µL PBS or PBS alone were injected via the tail vein into NSG and C57BL/KaLwRij mice, 8–12 week-old mice (both male and female). Myeloma cell engraftment was confirmed by third week bioluminescence imaging. The mouse with no signal until third week post-injection is removed from the study.

### 2.3. In Vivo Bioluminescence Imaging

Mice were imaged from first week to eighth week post-inoculation to assess tumor burden. Mice were anesthetized using isoflurane and imaged after 10 min of D-luciferin (1.5 mg/mouse, Perkin Elmer, Waltham, MA, USA) I.P. injection using an IVIS Imaging System 200 Series (Perkin Elmer, Waltham, MA, USA). All the images were acquired by auto exposure. Using Living Image 4.7.4 software, region of interest (ROI) was generated to cover whole body, and the total flux (*p*/s) was obtained.

### 2.4. Blood Sampling and Serum IgG2b Level Measurement

Blood samples (50 µL) were collected by retro-orbital bleeding and separated by centrifugation at 8000 rpm for 5 min. The plasma was transferred into separate vials and stored at −20 °C for the assay. According to the manufacturer’s instructions, serum IgG2 levels were measured using a mouse IgG2b Uncoated ELISA Kit (Invitrogen, Carlsbad, CA, USA).

### 2.5. BMD Measurement and Bone Histomorphometric Analyses on the Spine

When the mouse shows endpoint criteria, it is sacrificed and stored at −20 °C (>2 days). Once the mouse is thawed at Room Temperature, the vertebra on the spine is extracted from the mouse. The vertebrae’s bone mineral content and bone mineral density were measured using a PIXI-mus bone densitometer with on-board PIXI-mus software (G.E. Lunar, Madison, WI, USA) adjusted with body weight. Lumbar vertebrae (L1–L6) from the control and MM mice of NSG and C57/KaLwRij mice were scanned using a microCT40 (SCANCO Medical, Bassersdorf, Switzerland). For histomorphometric analyses, the trabecular bone regions were selected to assess the complete volume and structural appearances at each lumbar vertebrae. The analyses provided information regarding the main histomorphometry parameters such as bone volume (BV/TV, %), Trabecular thickness (Tb. Th, µm), number (Tb. N, n), and space (Tb. Sp, µm).

### 2.6. Statistical Analysis

Statistical analyses were conducted with a Student’s *t*-test. A *p* value of <0.05 was considered significant. Graph Prism7 (GraphPad Software Inc, San Diego, CA, USA) and SigmaPlot 13.0 (Systat Software Inc, San Jose, CA, USA) were used for all statistical analyses.

## 3. Results

### 3.1. 5TGM1-Luc Transplanted NSG Mouse Shows Early BL Signs, but Longer Survival than C57BL/KaLwRij Mouse

We have previously monitored MM progression in 5TGM1 intravenously transplanted C57BL/KaLwRij by evaluating serum immunoglobulin (Ig) changes and other phenotypic changes such as hindlimb paralysis, significant weight loss, and moribund state [17,18,19]. Both parameters become apparent at the terminal stage of MM progression.

For easy and early accessing of MM progression and MM onset in the mouse, we obtained the *Luciferase* gene-expressing 5TGM1 cells (5TGM1-*Luc*) from Dr. Oyajobi via collaboration and injected 1 × 10^6^ 5TGM1-*Luc* cells into C57BL/KaLwRij or NOD-*scid*IL2R^null^ (NSG; Jackson Laboratory, Bar Harbor, ME, USA) mouse via tail vein. Each week we performed in vivo bioluminescence (BL) imaging and collected blood (50 µL) for the serum Ig assay. We found the first signs of MM engraftment and growth at the second week of post-injection in NSG mice compared to the fourth week in C57BL/KaLwRij mice demonstrated by a focal signal on BL Image analyses (Figure 1A). As MM progressed, the BL signals increased and the focal BL positive areas spread throughout the body (Figure 1A). Like 5TGM1 transplanted C57BL/KaLwRij mice, the endpoint phenotypes (i.e., hindlimb paralysis, significant reductions of activity, and weight loss) were seen in NSG mice at either seventh or eighth-week post-transplantation. Engraftments of 5TGM1-*Luc* cells via tail vein in both C57BL/KaLwRij mice and NSG mice resulted in MM progression with typical MM phenotypes. Furthermore, we lost C57BL/KaLwRij mice in the fifth to seventh week of post-injection while NSG mice in the seventh to ninth week (Figure 1A). Comparison of the mouse survival by the log-rank (Mantel–Cox) test revealed that the survivals of the two models are significantly different (*p* = 0.0062) with 44.5 days in the C57BL/KaLwRij model versus 56 days of the median survival of the NSG model (Figure 1B).

### 3.2. Tumor Burdens Are Detected as Early as Week 2 of Post-Injection by BL Analysis

Since serum Ig analysis is a commonly used method in preclinical models to assess MM burden, we compared sensitivities of tumor burden by serum Ig and BL Imaging in both C57BL/KaLwRij and NSG mice. As mentioned in 3.1, all transplanted mice showed hindlimb paralysis, a typical sign of MM progression at the terminal stage.

For serum Ig analysis, the plasma were isolated from collected sera each week and stored at −20 °C as described in the Material and Method section. Serum IgG2b was measured by ELISA kit (Invitrogen Co., Carlsbad, CA, USA) Due to a difference in survival rate, serum collection ended at week 5 in C57BL/KaLwRij mice while continuing until week 6 in NSG mice. As expected, NSG has an undetectable level of IgG2b in control mice, so expressed as 0 mg/mL. As shown in Figure 2A, a significant increase of serum IgG2b level in NSG mice was noticed from 4th week and continuously increased in a ranges of 0.287–2.564 mg/mL. On the contrary, no significant change of IgG2b level in C57BL/KaLwRij mice were shown on weeks 3 and 4 with averages ranging between 0.267 and 0.363 mg/mL from 0.206 mg/mL at week 0. At week 5, serum IgG2b sharply increased to 1.303 mg/mL, suggesting that tumor burden in C57BL/KaLwRij mice become apparent, resulting in death. Apparent differences of serum Ig levels from the baseline appeared in the fourth to fifth week of post-injection in C57BL/KaLwRij mice while appearing in the third to sixth week in NSG mice.

For quantitative BL signals, we created a region of interest (ROI) box to cover the whole body and obtained total flux values (photons/sec) of each mouse from each week images (Figure 1A). The total flux values per mouse in the weekly images were plotted to express tumor burden (Figure 2B). In this graph, the tumor burdens were detected from the second week of post-injection and increased over time in NSG and C57BL/KaLwRij mice. All NSG mice showed relative uniformity of tumor burden at each timepoint than the C57BL/KaLwRij except first week. However, such variations in the C57BL/KaLwRij model may be partially due to a signal interruption from the black color and spots of the mouse.

These findings indicate that the NSG model has a better window to monitor the MM progression quantitatively.

### 3.3. Both Models Demonstrate Loss of Bone Mineral Using the Ex Vivo DEXA Scan

As 5TGM1 transplanted C57BL/KaLwRij mouse develops severe MMBD [35], we were interested to see how severe MMBD develops in these models. We measured the bone minerals in the spines of MM developing NSG or C57BL/KaLwRij mice compared to controls using PIXImus Densitometer (G.E. Lunar, Madison, WI, USA). At postmortem, we performed an ex vivo DEXA scan.

As shown in Figure 3, ex vivo DEXA analysis reveals that MM-induced NSG mice showed a significant reduction of the bone mineral density (BMD) (*p* = 0.0043) in the spine compared to their control. In contrast, MM-induced C57BL/KaLwRij mice showed a reduction of BMD recorded as (*p* = 0.0197) compared to control. Although differences of MM vs. control in both models demonstrated significant bone mineral loss, the more minor variation in the NSG model made a further significant difference than the C57BL/KaLwRij model.

### 3.4. More Severe MMBD Developed in NSG Mice Than C57BL/KaLwRij

Since severe bone mineral loss occurred in these models, we further evaluated the myeloma-induced bone lesions. After the DEXA scan, the spine and femur/tibia were extracted from a carcass. Thirteen spines from MM C57BL/KaLwRij mice and 12 spines from MM NSG mice with seven spines from control mice per strain were scanned by micro-computed tomography (microCT; Scanco Medical AG, Switzerland). For histomorphometry analyses, the trabecular regions on the lumbar vertebrae (L1–L6) were analyzed by drawing contour. The bone volume density (BV/TV) and trabecular thickness (Tb.Th) from whole vertebrae were expressed, but further significances in NSG showed than C57BL/KaLwRij (Figure 4A). In addition, the trabecular numbers and spaces were examined, but no statistical difference was seen.

The BV/TV and Tb.Th were significantly reduced in MM NSG mice from control mice (no-MM) (*p* < 0.0001), while these parameters were also considerably reduced (*p* = 0.005 and 0.009) in C57BL/KaLwRij mice (Figure 4A). The *p*-value differences come from variations of C57BL/KaLwRij since its standard errors were two to three folds higher than NSG mice. The percent reduction rates were calculated against the control and expressed in Figure 4B to compare each lumber. Both graphs showed unequal bone lesions in each lumbar. In the BV/TV, L2 of C57BL/KaLwRij and L2/3 of NSG showed higher reduction rates. In Figure 4C, the representative transverse microCT images and 3D-reconstructed images of the trabecular region in the vertebral body of corresponding lumbar (L3) from Control and MM mice are shown here. Although both images demonstrated significant bone loss on the trabecular region of the lumbar, 3D-reconstructed images showed further bone loss in the NSG model, in agreement with Figure 4B. These results demonstrate that both models induce significant bone loss, but the NSG model has more severe bone loss than C57BL/KaLwRij model.

## 4. Discussion

MMBD, a hallmark characteristic of MM, increases morbidity and mortality [36] and 60% of MM patients develop a fracture during the course of the disease [37]. While many MM animal models have been developed and enabled researchers to interrogate the mechanisms of MM tumorigenesis [12,16,38,39,40,41,42,43,44,45,46], the animal models that resemble human MMBD phenotypes were limited. Furthermore, several groups demonstrated that longitudinal assessments of MM tumor burden were limited to orthotopic transplantation models or insufficient numbers [22,47,48].

In the current study, we present an improved MMBD preclinical model with uniformity of tumor burden. This MM model was established by intra-venous engraftment of luciferase-transduced 5TGM1 MM cells into NSG. The tumor engraftment in this model was observed as early as the second week of post-injection using BL imaging and later confirmed by serum IgG2b surge. The BL imaging showed myeloma cell’s dissemination and growth patterns similar to human myeloma. In addition, this model demonstrated typical MM pathological phenotypes, such as hindlimb paralysis, significant weight loss, and moribund state. Compared to C57BL/KaLwRij model, we found that i) the 5TGM1–*Luc* engrafted NSG model survives longer; ii) tumor burdens are apparent from the control at an earlier timepoint; and iii) it showed a uniform tumor burden throughout the testing period.

Since the BL signals are interrupted by the color of the skin and fur, BL has not been used extensively in C57BL derived strains. Although we cream-shaved the back of C57BL/KaLwRij mice before BL imaging, the tumor burden variations may be partially contributed by the BL signal hindrance. However, our results clearly showed that the 5TGM1-Luc engrafted NSG model has slow MM progression and uniform tumor burden that benefit from evaluating novel myeloma therapeutics. Delaying MM progression in the NSG model from the C57BL/KaLwRij model demonstrated that the lack of immune cells causes this delay, supporting that the interactions of immune cells/MM cells/bone cells promote MM progression [10].

Human MMBD features bone lesions that can be found in various forms (i.e., classic discrete lytic lesion, diffused osteopenia, or multiple lytic lesions) at many parts of the skeleton, preferably spine, skull, and long bones [49]. However, many studies of MMBD were limited to the tibia and femur due to orthotopic injection [13,35,50,51,52,53] or the vertebral bone lesion study was not thoroughly evaluated [54]. Furthermore, human MMBD causes several complications (i.e., bone pain, fractures, hypercalcemia, and spinal cord compression), resulting in decreased quality of life and poor mobility [55]. These bone lesions in MMBD are always lytic and promote the processes of bone resorption. The mechanism associated is the displacement of the RANKL/OPG ratio, which affects the process of osteoclastogenesis [10,56]. Another exciting feature in 5TGM1-*Luc* engrafted NSG model is severe osteolytic bone lesions of the lumbar spine. In ex vivo DEXA analysis, we found a significant reduction of BMD in the spine of NSG mice from non-MM mice (*p* = 0.0043), whereas a less significant change in the C57BL/KaLwRij model (*p* = 0.0197). In microCT analyses, both models (NSG and C57BL/KaLwRij) show substantial bone loss (BV/TV and Th.Th) compared to non-MM mice. However, the NSG model showed further significance. These findings aim that the osteolytic lesions observed in our study may be due to aberrant RANK/RANKL signaling along with subsequent failure of immune system in MM [10] which needs to be studied and explored in the further study.

Our results here provide first insights into quantitative data of the myeloma progression in the NSG mouse after systemic 5TGM1–*Luc* injection and qualitative data of the myeloma-induced bone lesions, as proof of concept for detecting advanced ultrastructural deviations in the spine. These quantitative and qualitative assessments of MMBD will be useful to test various anti-MMBD drugs.

## 5. Conclusions

We improved the MM animal model which recapitulates human myeloma with secretion of paraprotein, lytic bone lesions and spinal compression. Our newly developed NSG model may allow an investigator to characterize various stages of the MMBD down to the ultrastructural level and serve as a valuable model to test multiple anti-MMBD drugs for their efficacy.

## Figures and Tables

**Figure 1 cancers-13-04277-f001:**
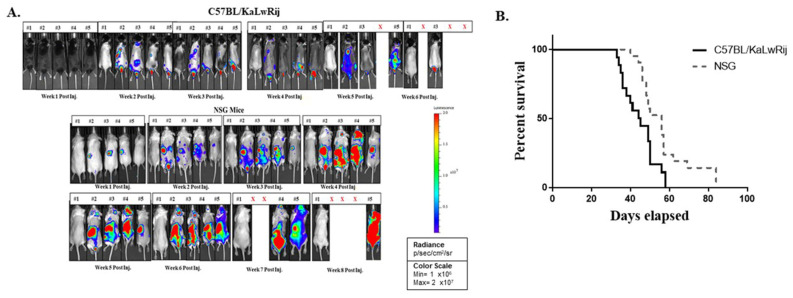
Differences of myeloma development in C57BL/KaLwRij and NSG models. (**A**) Representative weekly BL images of 5TGM1-Luc transplanted C57BL/KaLwRij and NSG Models. All BL signals were normalized in one scale as shown. 1 × 10^6^ 5TGM1-Luc cells in 100 ul of PBS or PBS alone were injected to C57BL/KaLwRij or NSG mice (equal genders in 8~12 weeks old age) via the tail vein. In each group, the first one (#1) is a control whereas #2~#5 are 5TGM1-Luc injected mice. (**B**) Mice were monitored weekly. When the mouse meets endpoint criteria, it is sacrificed and recorded. Survival curves representing 18 mice from C57BL/KaLwRij (solid line) and 21 Mice from NSG (dotted line). Median survival was 56 days for NSG compared to 44.5 days for C57BL/KaLwRij. The Mantel–Cox test showed *p* = 0.0062. X denotes mouse died during that week.

**Figure 2 cancers-13-04277-f002:**
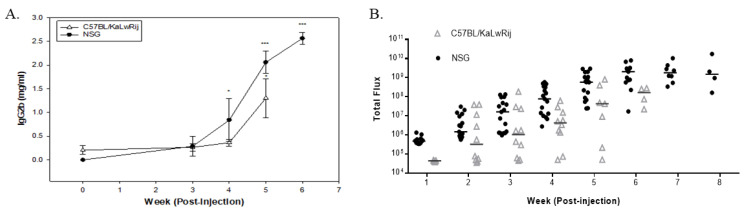
Monitoring myeloma progression using two different parameters; serum IgG2b levels and whole-body bioluminescence signals. 1 × 10^6^ 5TGM1-Luc cells in 100 ul of PBS or PBS alone were injected to C57BL/KaLwRij or NSG mice (equal genders in 8~12 weeks old age) via the tail vein. Every week BL imaging was taken and blood were drawn. (**A**) IgG2b were measured from plasma using ELISA kit. Mean serum IgG2b levels were calculated and expressed from NSG control mice and C57BL/KaLwRij. All undetectable IgG2b levels were brought to 0 mg/mL. Data are expressed as mean ± S.E. * denotes *p* < 0.05, *** denotes *p* < 0.001 compared with their baseline. (**B**) From weekly image, total flux values were obtained from the regions of interest covering the whole body of mouse. All total flux values (closed circle or open triangle) with median (line) are expressed as a progression of tumor burden.

**Figure 3 cancers-13-04277-f003:**
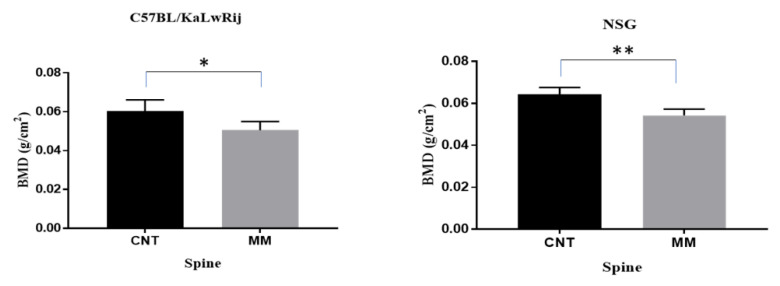
Bone mineral density changes in 5TGM1 bearing mice versus the control mice. The mouse was sacrificed and kept in the freezer for >2 days. Once the carcass was thawed, the vertebra was extracted from the frozen carcass and scanned using PIXI-mus. BMD values are expressed here from control versus 5TGM1 bearing mice per each model and tested using Student’s *t*-tests. * denotes *p* < 0.05, ** denotes *p* < 0.01.

**Figure 4 cancers-13-04277-f004:**
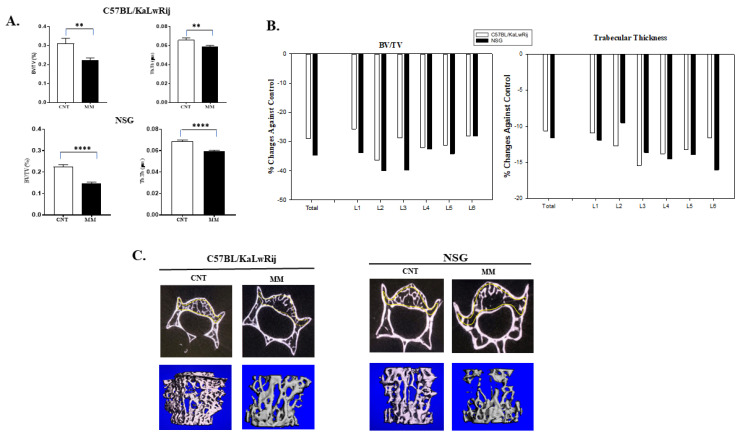
Myeloma associated bone lesions were assessed by microCT analysis. (**A**) The trabecular regions at total lumbar vertebrae (L1–L6) were analyzed and expressed changes of bone volume (BV/TV) and trabecular thickness (Tb.Th) in each models (NSG and C57/KaLwRij) as control vs MM. BV/TV and Tb.Th. values are expressed here from control versus 5TGM1 bearing mice in each model and tested using Student’s *t*-tests. ** denotes *p* < 0.01, **** denotes *p* < 0.001. (**B**) Percentages of change of BV/TV and Tb.Th. were calculated by [(Mean of Control–Mean of MM)/Mean of Control × 100] and expressed. Total vertebrae or each lumbers were expressed. (**C**) Representative 2D Transverse microCT images and 3D reconstructed images were shown here. All transverse pictures were shown here at the same distance from the growth plate of L3 of each mouse. 3D images were obtained from reconstructed of trabecular region in L3 of each mouse.

## Data Availability

The data presented in this study are available upon request from the corresponding author.

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
