# Peer review of "An Improved Animal Model of Multiple Myeloma Bone Disease"

_cancers, 2021, doi:10.3390/cancers13174277_

Round 1

Reviewer 1 Report

Syed Hassan Mehdi et al uncovered a novel Animal Model of Multiple Myeloma Bone Disease. Point to be cosidered:

  1. Mice model: when discussing the methodology applied for number selection it is not clear for my understanding, how the author selected the sufficient experimental setting. Indeed, in the in vivo experiments, sample size has been should be calculated in a rigorous way, i.e. by using G*Power software (power of for example 80% and 0.05 statistical level, etc.). Assuming an effect-size of for example, 0.4 with statistical significance of α <;0.05 and a power of 80%. Can the authors comment on this topic?
  2. why day 56 is reported as obseravtion bowndary for survival, did the author followed the animals longer, what about spinal engraftment?
  3.  C57BL/KalwRij are prone to develop B cell lymphoproliferative disorders, as approximately 80% of these mice carry a monoclonal component (MC), resembling human MGUS. A very small fraction (0.5%) of mice progress to MM and Waldenstrom Macroglobulinemia. Mice spontaneously developing MM disease represent the original 5TMM model. Bone marrow (BM)-MM cells from these mice can be efficiently transplanted into syngeneic mice to easily reproduce the disease. Indeed, from the original 5TMM, several cell lines have been established, such as: 5T2 that reproduces different milestones of advanced disease including serum paraprotein production and lytic bone lesion formation; 5T33, more aggressive, with preferential dissemination to spleen and liver. It is therefore possible to derive mice carrying extensive or very limited skeleton damages, with malignant PC clones confined to the BM and spleen and with different growth patterns. In this model, cytogenetic abnormalities showed hyperdiploid features, with lower frequency of translocations as compared to human disease. Can the authors comment on this?
  4. The recently introduced NOD/SCID gamma null (NSG) mouse strain lacks IL-2 gamma chain, resulting in the absence of B/T/NK activity and severe impairment of antigen presenting cells and complement. Some authors attained successful engraftment of I.V. injected U266 MM cells into NSG recipients following 2.4 gy irradiation (demonstrated to be unnecessary for tumor engraftment in subsequent reports). U266 showed specific homing to murine BM, which is observed also with other MM cells such as OPM2, JJN3 and RPMI8226, and in contrast to NOD/SCID IV implants (PMID: 32354870). Homing to the BM is reproducibly attained and classical features of MMBD can be observed with increased OCL activity, OBL reduction and lytic lesions across the skeleton. However, it should be taken into account, that these models may present an extensive burden of extramedullary disease, thus mostly resembling very aggressive or late stage MM disease more than classical indolent myeloma. Again, the major limitation of this model is due to cell lines, that are better representative of a BMM-independent plasma cell leukemia than a MM. Can the authors comment?
  5. This reviewer personally misses some important implications that can boost the authors'findings while boosting the interest for a broad readership in the oncology community: MM is often associated with adverse SREs. The bone lesions in MM are always lytic and it depends on their ability to promote the processes of bone resorption. The principal mechanism is the displacement of the RANKL/OPG ratio to the process of osteoclastogenesis. Collectively, Osteoimmunology was coined about twenty years ago to identify a strict cross talk between bone niche and immune system both in physiological and pathological activities, including cancer. Several molecules are involved in the complex interaction between bone niche, immune and cancer cells. The Receptor Activator of NF-kB (RANK)/RANK Ligand (RANKL/Osteoprotegerin (OPG) pathway plays a crucial role in bone cells/cancer interactions with subsequently immune system control failure, bone destruction, inhibition of effect, and metastasis outcome. The bidirectional cross-talk between bone and immune system could become a potential target for anticancer drugs (PMID: 32064051). Can the author discuss and expand introduction and discussion accordingly?

Author Response

  1. Mice model: when discussing the methodology applied for number selection it is not clear for my understanding, how the author selected the sufficient experimental setting. Indeed, in the in vivo experiments, sample size has been should be calculated in a rigorous way, i.e. by using G*Power software (power of for example 80% and 0.05 statistical level, etc.). Assuming an effect-size of for example, 0.4 with statistical significance of α <;0.05 and a power of 80%. Can the authors comment on this topic?

Power calculation for the two-group survival analysis suggests we might need a sample size of at least 45 for each group to achieve 0.8 statistical power, when the proportion of subjects in group is 0.5, relative hazard is 2.9, baseline event rate is 0.2 per day, median survival time in Group 0 is 3.466. The type I error (alpha-level) is set to 0.05. Note this calculation is done based on our current study, and applying this idea is only prospective, i.e., we can only apply the knowledge for future experiment. However, it may still be meaningful to interpret our current study. Our current sample size (18, 21 each group) is not necessarily sufficient, but it shows already the significance. With increased sample size in the future, we are likely to have more solid same conclusion. We use the tool provided at https://sample-size.net.

2. why day 56 is reported as obseravtion bowndary for survival, did the author followed the animals longer, what about spinal engraftment?

56 days after a transplant was the median days for MM bearing NSG. The study carried up to 85 days when the last MM bearing mouse died. Figure 1 showed spinal engraftment at early stage of progression, but MM cells spread whole body at the terminal stage.

3.  C57BL/KalwRij are prone to develop B cell lymphoproliferative disorders, as approximately 80% of these mice carry a monoclonal component (MC), resembling human MGUS. A very small fraction (0.5%) of mice progress to MM and Waldenstrom Macroglobulinemia. Mice spontaneously developing MM disease represent the original 5TMM model. Bone marrow (BM)-MM cells from these mice can be efficiently transplanted into syngeneic mice to easily reproduce the disease. Indeed, from the original 5TMM, several cell lines have been established, such as: 5T2 that reproduces different milestones of advanced disease including serum paraprotein production and lytic bone lesion formation; 5T33, more aggressive, with preferential dissemination to spleen and liver. It is therefore possible to derive mice carrying extensive or very limited skeleton damages, with malignant PC clones confined to the BM and spleen and with different growth patterns. In this model, cytogenetic abnormalities showed hyperdiploid features, with lower frequency of translocations as compared to human disease. Can the authors comment on this?

5TGM1 cells is one of subsequent subclones derived from the 5T33MM model.  These clones (the 5T33vt and 5TGM1 cells) were clonally related in vitro growing cell lines, stroma independent.   It has been used to investigate the mechanisms underlying MM-induced bone disease.  The cytogenetic study showed that compared to 5T2 the 5TGM1 carries less chromosomal aberrations, but shares many common key alterations.  Despite some variation of the 5T33 and 5TGM1 cells, they share many genetic aberrations such as del13q in patients and a mutation in Trp53 (Maes et al, 2018).  We aware that the 5TGM1 model represent a subset and the advanced stages of MM disease.  The 5TGM1 is the single clone and we transplanted the same cells simultaneously into both C57BL/KaLwRij and NSG mice.  We consider that variations of bone lesions and tumor burdens are due to difference of the mouse strains rather than different clones.

As well as tumor burden (Garett et al., 1997), the 5TGM1 bearing C57BL/KaLwRij model has been widely used in the field of myeloma bone disease, due to their pronounced osteolytic lesion formation (Asosingh et al, 2000).  Lawson et al, 2015 demonstrated that this model represents the dormant stage of MM cells and the 5TGM1 producing extracellular vesicles induced osteolysis of trabecular bone volume (Faict et al, 2018).  We have been studying the 5TGM1 bearing C57BL/KaLwRij model (Zangari, et al, 2014, 2018, and 2018, Bone, Endocrinology, and Journal of Bone Oncology).  As we found the limits of the current 5TGM1 bearing C57BL/KaLwRij model, we tried the 5TGM1 transplanting NSG model and found that this model demonstrates better for assessment of tumor burden and bone lesions as shown in this manuscript.

4. The recently introduced NOD/SCID gamma null (NSG) mouse strain lacks IL-2 gamma chain, resulting in the absence of B/T/NK activity and severe impairment of antigen presenting cells and complement. Some authors attained successful engraftment of I.V. injected U266 MM cells into NSG recipients following 2.4 gy irradiation (demonstrated to be unnecessary for tumor engraftment in subsequent reports). U266 showed specific homing to murine BM, which is observed also with other MM cells such as OPM2, JJN3 and RPMI8226, and in contrast to NOD/SCID IV implants (PMID: 32354870). Homing to the BM is reproducibly attained and classical features of MMBD can be observed with increased OCL activity, OBL reduction and lytic lesions across the skeleton. However, it should be taken into account, that these models may present an extensive burden of extramedullary disease, thus mostly resembling very aggressive or late stage MM disease more than classical indolent myeloma. Again, the major limitation of this model is due to cell lines, that are better representative of a BMM-independent plasma cell leukemia than a MM. Can the authors comment?

We appreciate reviewer’s thoughtful comment in this matter.  We agree with the reviewer’s opinion.  Previous works by other groups including Swift et al 2012 and Lawson et al, 2015 demonstrated that IV injection of various human myeloma cells infiltrated murine BM.  However, the works, we saw, either did not evaluate bone lesions or demonstrated variations and evaluated the tibiae of the transplanted mice.  On our models, we discovered that the bone lesions at the tibiae are varied and could not reach to a statistically significant difference compared to non-myeloma mouse group.  We agree that the 5TGM1 model represent a BMM-independent PCL.  However, it resembles many features of MMBD and this model can provide a starting point to test MMBD targeted therapy under the current circumstance (no good model for a MMBD study). 

Besides, we are working on developing MM models to represent the earlier stage of MM disease using humanized mouse with patient-derived MM cells.

5. This reviewer personally misses some important implications that can boost the authors' findings while boosting the interest for a broad readership in the oncology community: MM is often associated with adverse SREs. The bone lesions in MM are always lytic and it depends on their ability to promote the processes of bone resorption. The principal mechanism is the displacement of the RANKL/OPG ratio to the process of osteoclastogenesis. Collectively, Osteoimmunology was coined about twenty years ago to identify a strict cross talk between bone niche and immune system both in physiological and pathological activities, including cancer. Several molecules are involved in the complex interaction between bone niche, immune and cancer cells. The Receptor Activator of NF-kB (RANK)/RANK Ligand (RANKL/Osteoprotegerin (OPG) pathway plays a crucial role in bone cells/cancer interactions with subsequently immune system control failure, bone destruction, inhibition of effect, and metastasis outcome. The bidirectional cross-talk between bone and immune system could become a potential target for anticancer drugs (PMID: 32064051). Can the author discuss and expand introduction and discussion accordingly?

We appreciate reviewer’s excellent comments. It clearly addresses what we missed in this manuscript. We added the sentence in the sections of Introduction and Discussion. Modifications were highlighted and can be found in the lines 43-48, 268-271, 278-281, and 287-290.

Reviewer 2 Report

The authors injected 5TGM1 cell lines to NSG mice via tail vein to setup animal model of myeloma disease. With this model, the authors presented an improvement of animal model on myeloma bone disease than previous models. The study is well designed and the manuscript is well prepared.

  1. Since the model seems to have a better presentation on bone lesions, how about the biomakers of bone lesions in this model and the compared model?
  2. How about other presenations of myeloma, such as anemia, hypercalcemia, renal function, in this model?

Author Response

1. Since the model seems to have a better presentation on bone lesions, how about the biomakers of bone lesions in this model and the compared model?

We appreciate comment from the respected reviewer.  We found hypercalcemia in both MM models, but can’t access other biomarkers (such CTX-I or TRAP-5) due to limited serum available at present.  We regret to save sufficient blood for all these biomarker analyses.  The study will be designed for the future study.

2. How about other presenations of myeloma, such as anemia, hypercalcemia, renal function, in this model?

We appreciate the suggestion.  We have found anemia on some NSG models, but not in C57BL/KaLwRij model.  As mentioned above, hypercalcemia was found in both MM models.  All these analyses need more experiments to confirm the findings.  We will design better for future studies.

Round 2

Reviewer 1 Report

The authors corrected the manuscript according to the previous comments.